# Framing and Management of Migraines in Women: An Expert Opinion on Challenges, Current Approaches, and Future Multidisciplinary Perspectives

**DOI:** 10.3390/healthcare13020164

**Published:** 2025-01-16

**Authors:** Piero Barbanti, Rossella E. Nappi

**Affiliations:** 1Headache and Pain Unit, IRCCS San Raffaele, 00163 Rome, Italy; piero.barbanti@sanraffaele.it; 2San Raffaele University, 00166 Rome, Italy; 3Department of Clinical, Surgical, Diagnostic and Pediatric Sciences, University of Pavia, 27100 Pavia, Italy; 4Research Center for Reproductive Medicine, Gynecological Endocrinology and Menopause, IRCCS S. Matteo Foundation, 27100 Pavia, Italy

**Keywords:** migraines in women, hormonal fluctuations, CGRP-targeted therapies, reproductive milestones, personalized migraine management, technological innovations in migraine care

## Abstract

**Background/Objectives:** Migraines are a common neurological disorder that significantly impact women, especially during their reproductive years. Hormonal, neurological, and lifestyle factors shape migraine patterns, with fluctuations during menstruation, pregnancy, perimenopause, and menopause influencing migraine prevalence and severity. This expert opinion explores current challenges, therapeutic strategies, and future directions for personalized care, addressing the limited inclusion of women in clinical research across different life stages. **Methods:** In order to focus on hormonal influences, pharmacological and non-pharmacological therapies, including CGRP monoclonal antibodies, neuromodulation, and lifestyle interventions, a comprehensive analysis of literature, in particular on clinical trials, real-world studies, and guidelines on migraine management was performed. Emerging digital tools and AI-based approaches were also evaluated to improve personalized care for women with migraine. **Results:** Hormonal therapies, including contraceptives and HRTs, present both risks and benefits, particularly for women with migraines with aura, highlighting the need for individualized approaches. Advances in CGRP-targeted therapies have shown effectiveness in preventing refractory migraines. Non-pharmacological treatments, such as neuromodulation, acupuncture, and lifestyle adjustments, further expand the treatment landscape. However, research gaps remain, particularly regarding hormonal influences on migraines during pregnancy and menopause. **Conclusions:** Future research should prioritize female-specific clinical trials to better understand the impact of hormonal changes on migraines. Tailored therapies combining pharmacological, non-pharmacological, and digital solutions are essential for improving care. A multidisciplinary approach integrating personalized medicine, technological advancements, and patient education is crucial to optimizing outcomes and enhancing quality of life for women with migraine.

## 1. Introduction

Migraines are a prevalent neurological disorder disproportionately affecting women, especially during their reproductive years. Globally, it is the leading cause of disability in women aged 15–49, severely impacting quality of life, work productivity, and social interactions [1,2]. The burden of migraines is exacerbated by their complex pathophysiology, which involves thalamo-cortical dysrhythmia, central sensitization, and a lack of habituation to sensory stimuli. These mechanisms contribute to the cyclic nature of a migraine attack, which progresses through distinct phases, including prodrome, aura, headache, and postdrome, further complicating diagnosis and management [3,4]. Migraine prevalence varies across life stages due to hormonal changes, peaking during puberty, childbearing years, and perimenopause [5,6]. Recognizing these gender-specific factors is essential for advancing diagnostic and therapeutic strategies tailored to women’s needs.

Migraines affect over one billion people worldwide, with women being three times more likely than men to experience them after puberty [7,8]. This gender disparity is closely tied to hormonal fluctuations, as migraine onset often coincides with menarche and continues through reproductive years, pregnancy, and menopause [6,9]. Hormonal contraception offers potential benefits for managing migraines linked to estrogen withdrawal. Progestin-only contraceptives stabilize hormone levels and reduce cortical excitability, making them effective for managing migraines with auras [10]. Natural estrogen-based formulations, developed with bioidentical estrogens, provide safer alternatives to synthetic estrogen products, particularly during menopausal transitions [10,11].

Women experience more frequent, more severe, and longer-lasting attacks than men, contributing to a greater disease burden [2]. Epidemiological studies highlight peaks in prevalence during reproductive age [12,13]. The highest prevalence occurs between ages 35 and 39, reflecting cyclic hormonal changes during the menstrual cycle [14]. Menstrual migraines affect up to 60% of women with migraines, underscoring the importance of addressing hormonal cycles in their management [9,15].

Hormonal changes across life stages further illustrate estrogen’s role in migraine pathophysiology. Pregnancy, characterized by stable, high estrogen levels during the second and third trimesters, often reduces migraine frequency. However, the postpartum period, marked by a rapid estrogen decline, is associated with worsened symptoms [2,14,16]. As highlighted in recent reviews, estrogen’s impact on migraine varies with the timing, dosage, and delivery method, emphasizing the importance of understanding these dynamics for effective management strategies [17]. Perimenopause, with erratic hormonal fluctuations and shorter menstrual cycles, frequently exacerbates a migraine’s severity, particularly in women with menstrual migraines [18].

In menopause, migraines without auras generally decline, though vasomotor symptoms (VMSs) such as hot flashes and night sweats are more common among women with migraines. These shared mechanisms, including vascular and autonomic dysregulation, emphasize the need to address both conditions to mitigate cardiovascular risks [18,19]. Continuous hormonal regimens, including transdermal estrogens, stabilize hormone levels and may reduce migraine episodes, though long-term studies are needed to confirm their safety and efficacy [18]. The choice of hormonal therapy, including its composition and route of administration, can significantly influence migraine patterns, as exogenous estrogen and progestogen interact differently with underlying neurovascular mechanisms [17].

Emerging therapeutic strategies, including hormone-based approaches, offer promise for improving migraine management across life stages. Further research is essential to refine personalized treatments for women at different reproductive phases [7,20].

### 1.1. Expert Opinion Objectives

This expert opinion seeks to highlight key points of novelty regarding the complex interplay between hormonal changes and migraines in women, emphasizing the importance of personalized treatment approaches. Given the emerging landscape of migraine therapies, including the use of calcitonin gene-related peptide (CGRP) inhibitors and hormonal treatments, how these therapies can be integrated to optimize care for females with migraines will be explored. The goal is to inform clinical decision making, address gaps in current treatment paradigms, and promote multidisciplinary approaches which integrate insights from neurology, gynecological endocrinology, and primary care [21,22]. This expert opinion then also seeks to raise awareness about the impact of migraines on women, encourage further research into gender-specific mechanisms of the disease and promote the development of effective, individualized treatment pathways which can enhance the quality of life for female migraine sufferers worldwide.

### 1.2. Burden of Disease in Women

The condition’s debilitating nature contributes to significant economic costs, with affected individuals facing higher healthcare expenses, increased absenteeism, and reduced productivity at work [1,23]. Many women report difficulties maintaining consistent work performance due to frequent migraine attacks, leading to both absenteeism and presenteeism, where they are present at work but unable to perform effectively [24,25]. Additionally, the post-headache phase, often referred to as the postdrome, further impacts daily functioning. Many patients experience lingering symptoms, such as fatigue, mood changes, and cognitive difficulties, which can significantly impair productivity and quality of life even after the pain subsides. Effective treatments which address both the headache and associated symptoms are reported to provide substantial relief and improved functioning [8].

Furthermore, migraines interfere with social and family responsibilities, adding to psychological stress and reducing the overall quality of life [8,15]. Despite the significant burden, migraines remain underdiagnosed or misdiagnosed and undertreated, particularly in women, due to variability in symptom presentation and the complex interplay of hormonal factors [20]. A recent study highlighted the overuse of triptans among women, underscoring a need for improved management and preventive strategies [13].

## 2. Migraines Across Different Stages of a Woman’s Life

### 2.1. Menstrual Migraine

The menstrual migraine is a subtype of migraines which occurs in close association with the menstrual cycle, typically triggered by a rapid decline in estrogen levels during the late luteal phase. It is characterized by more severe, longer-lasting, and treatment-resistant attacks compared with non-menstrual migraines [15,26]. According to epidemiological studies, up to 60% of female migraine sufferers reported a link between migraine attacks and their menstrual cycles, highlighting the significant role of hormonal fluctuations [12]. Menstrual migraines can be categorized as either pure menstrual migraines (PMMs), which occur exclusively around the menstrual period, or menstrually related migraines (MRMs), which occur during menstruation but also at other times of the month [15]. Treatments for menstrual migraines often involve short-term prophylactic therapies, such as the use of triptans, initiated a few days before the expected onset of menstruation, as well as nonsteroidal anti-inflammatory drugs (NSAIDs) to mitigate pain during attacks (Table 1) [5,15]. The decline in estrogen levels during the perimenstrual period is recognized as a primary trigger for menstrual migraine attacks, which can be more severe and resistant to standard treatments [27]. Short-term prophylactic treatments, such as triptans and nonsteroidal anti-inflammatory drugs (NSAIDs), have demonstrated varying degrees of efficacy in managing menstrual migraines (Table 1). While the first class of drugs has shown significant reductions in migraine frequency and severity, the second class’s effects are less consistent, offering limited relief in some cases [26,27,28,29]. Further research is needed to determine whether extending the duration of estroprogestinic therapy or adjusting dosing strategies could minimize these post-treatment effects of estrogen withdrawal and enhance the overall outcomes.

For patients unresponsive to standard therapies, newer treatments are emerging. These include CGRP antagonists and oxytocin agonists, which specifically target mechanisms related to hormonal fluctuations, offering potential relief tailored to the unique characteristics of menstrual migraines (Table 1) [30].

**Table 1 healthcare-13-00164-t001:** Summary of key treatments and findings from clinical trials on menstrual migraine management.

	Treatment	Description	Dosage or Administration	Key Findings	References
Triptans	Frovatriptan	Serotonin receptor agonist	2.5 mg once or twice daily, starting 2 days before menstruation and continuing for 6 days	Significant reduction in headache occurrence during the perimenstrual period	Adelman and Calhoun (2005) [31].
	Naratriptan	Serotonin receptor agonist	1 mg twice daily 2–3 days before menses onset for 5–7 days	Reduced frequency and severity of menstrual migraines	Newman et al. 2001 [32]
	Zolmitriptan	Serotonin receptor agonist	2.5 mg twice daily 2–3 days before menses onset for 5–7 days	Similar efficacy to naratriptan in menstrual migraine prevention	Tuchman et al. 2008 [33]
NSAIDs	Naproxen sodium	Anti-inflammatory medication	550 mg twice daily 1–2 days before menstruation through the menstrual period	Moderate reduction in migraine frequency and severity; less consistent results than triptans	Allais et al. 2007 [34]
Hormonal Therapies	Percutaneous estrogen	Hormonal supplementation	1.5 mg gel, starting 2 days before menstruation for 7 days	Reduction in migraine days but potential post-treatment rebound due to estrogen withdrawal	Silberstein and Hutchinson 2008 [35]
CGRP Antagonists	Eptinezumab	Monoclonal antibody targeting CGRP	Intravenous administration	Reduction in migraine days; further research required for long-term safety and efficacy	Lipton et al. 2020 [36]

### 2.2. Migraines and Hormonal Contraception

Hormonal contraception can have variable effects on migraines, which are influenced by the type of contraception used and the individual’s migraine history. Combined hormonal contraceptives (CHCs), which contain both estrogen and progestin, may stabilize hormonal fluctuations and potentially reduce the frequency of menstrual migraines in some women by maintaining steady estrogen levels [26]. However, the hormone-free interval in traditional combined contraceptive regimens can lead to “estrogen withdrawal headaches”, triggering migraines during this period [13]. Continuous or extended-cycle hormonal contraceptive regimens, which eliminate or shorten the hormone-free interval, have been recommended as alternative strategies, potentially reducing migraine frequency and severity by maintaining stable hormone levels and preventing withdrawal effects [37]. Importantly, women with migraines with auras are advised to exercise caution when using CHCs, as they may have an increased risk of ischemic stroke, especially if additional risk factors like smoking or hypertension are present [15,22]. Furthermore, CHCs can induce the onset of auras in women who have not experienced this symptom in the past [38]. For this reason, the presence of an aura is considered an absolute contraindication for CHC use to mitigate stroke risk. In these cases, progestin-only contraceptives or non-hormonal methods may be recommended to minimize risk [26].

### 2.3. Migraines in Pregnancy and Lactation

Pregnancy significantly alters the hormonal milieu, leading to varied effects on migraine patterns. For many women, the migraine frequency decreases, especially during the second and third trimesters, due to stable, high levels of estrogen. Migraines without auras (MOs) generally improve during pregnancy, particularly in the second and third trimesters, as the hyperestrogenic state suppresses hormonal fluctuations [39]. In contrast, migraines with auras (MAs) exhibit distinct patterns, with some women experiencing persistence or even new onset of aura symptoms during gestation. This distinction is critical as MAs have been linked to an increased risk of pregnancy complications, including gestational hypertension, preeclampsia, and peripartum stroke [39].

Management of migraines during pregnancy is challenging, as many standard treatments, including triptans and NSAIDs, are either contraindicated or require cautious use due to potential risks to the fetus [40]. Non-pharmacological approaches, such as acupuncture, relaxation techniques, and lifestyle modifications, are often recommended as first-line options during pregnancy to minimize medication use [5,14]. However, in select cases, pharmacological prophylaxis may be considered, and serotonin receptor agonists, such as triptans, are regarded as safe options for managing acute migraine attacks during pregnancy [41,42]. In the postpartum period, the rapid decline in estrogen levels may trigger a resurgence of migraine attacks. During lactation, some analgesics, such as NSAIDs, are considered safer options, but the choice of therapy must always be guided by a careful assessment of the risk-benefit profile [14,21]. Breastfeeding may exert a protective effect against migraine recurrence, likely due to suppressed ovulatory cycles and stabilized hormone levels, further emphasizing its benefits for postpartum women [39].

### 2.4. Menopause and Postmenopausal Migraines

The transition into menopause, marked by fluctuating and declining estrogen levels, can alter migraine patterns. Perimenopause often leads to increased frequency and severity of migraine attacks due to hormonal instability [12,26]. Upon reaching menopause, migraine patterns may stabilize or improve, particularly for those with MOs [7].

Hormone replacement therapy (HRT) can offer benefits, alleviate vasomotor symptoms, and reduce estrogen withdrawal migraines, especially in women without auras. Continuous progestogen use is essential for endometrial protection and avoids exacerbating migraines. Non-hormonal options, including SSRIs and SNRIs, are effective alternatives for managing migraines and vasomotor symptoms when HRT is contraindicated [43].

HRT use can complicate migraine management. While some women experience relief with continuous low-dose estrogen therapy, others, especially those with a history of MAs, may see symptoms worsen [9,13]. Estrogen-sensitive neurons in the hypothalamus, which are crucial for regulating energy expenditure and activity, diminish in function after menopause, potentially influencing migraine severity and metabolic changes [44]. Tailoring HRT regimens to individual migraine profiles, including dose and delivery method, is key to minimizing risks and optimizing outcomes [21,26].

## 3. Challenges in Diagnosis and Management

### 3.1. Diagnostic Gaps and Stigmatization

Despite its prevalence and disabling nature, migraines remain underdiagnosed, misdiagnosed, and undertreated, particularly in women. A key diagnostic challenge is the variability in symptom presentation, which ranges from classic headaches with auras to non-specific symptoms such as dizziness, neck pain, and light sensitivity [24]. Auras, though often considered defining, occur in a minority of cases. Typical migraines feature unilateral, pulsating pain with vegetative symptoms like nausea, vomiting, photophobia, and phonophobia, aiding diagnosis. However, atypical presentations, such as bilateral, constrictive pain with cervical tenderness, often lead to diagnostic errors. The lack of objective biomarkers further contributes to delays and misclassification, particularly among women who may not recognize their symptoms as migraines [20].

Menstrual migraines, closely linked to hormonal fluctuations, are often mistaken for premenstrual syndrome (PMS), complicating accurate diagnosis [26]. Women with MRMs may attribute symptoms solely to hormonal imbalances, overlooking the central neurological origin of migraines. This perception fosters a sense of inevitability and delays access to targeted therapies, underscoring the need for education on migraines’ neurological basis to encourage timely treatment.

Stigmatization also exacerbates diagnostic challenges. Women with chronic migraines face skepticism from healthcare providers, colleagues, and family members, who may dismiss symptoms as exaggerated or excuses to avoid responsibilities [1,15]. This stigma reduces help-seeking and adherence, worsening outcomes. Additionally, viewing migraines as a “women’s problem” compounds these issues, leaving women with higher disability levels yet less effective treatment compared with men [12,13]. Tackling these diagnostic gaps and the stigma is vital for improving care access and outcomes.

### 3.2. General Practitioner’s Role

General practitioners (GPs) play a crucial role in the initial identification and management of migraines, yet many experience challenges in accurately diagnosing and assessing the severity of cases. Studies suggest that expanded training in migraine recognition and the use of standardized diagnostic tools, such as the ID-migraine, HIT-6, or MIDAS scales, could help GPs more effectively categorize migraine severity and identify high-risk patients [45]. Improved GP awareness and diagnostic accuracy are essential for reducing delays in treatment and ensuring timely referrals when specialized care is necessary [1].

### 3.3. Individualized Care and Multidisciplinary Approach

Given the complex interplay of hormonal, neurological, and lifestyle factors in migraine pathophysiology, a one-size-fits-all approach to treatment is often ineffective. Individualized care which considers a woman’s age, reproductive stage, comorbidities, and personal preferences is essential for effective migraine management [22]. Recent studies highlighted that patient preferences are often influenced by the need for effective relief with minimal side effects, such as avoiding cognitive impairment or weight gain. Many patients are willing to trade some efficacy for better tolerability, underscoring the importance of patient-centered approaches which align treatment choices with individual preferences. This approach not only improves adherence but also fosters greater patient satisfaction with care [46]. For example, menstrual migraines may benefit from tailored treatment strategies which include short-term hormonal prophylaxis, while women with migraines in pregnancy require non-pharmacological approaches to minimize risks to the fetus [14,15]. Hormonal therapies, including hormonal contraceptives and HRT, should be considered carefully, particularly in women with MAs, due to the potential risk of stroke [22,26].

A multidisciplinary approach ensures that the diverse factors influencing migraines, including hormonal, neurological, and psychosocial aspects, are addressed holistically. For example, neurologists provide expertise in diagnosing migraine subtypes and selecting appropriate pharmacological treatments, while gynecologists and endocrinologists can evaluate and manage hormonal influences, such as those related to menstruation, pregnancy, or menopause. Pain specialists contribute advanced management strategies for refractory cases, including neuromodulation and interventional techniques. Additionally, mental health professionals play a critical role in addressing anxiety, depression, and other comorbidities closely associated with chronic migraines. Integrated care models, where specialists work collaboratively in structured teams, have shown promise in improving patient outcomes. These models facilitate streamlined communication between providers, ensuring that care plans are tailored to individual needs.

This is particularly relevant for managing menstrual and menopausal migraines, where hormonal factors play a significant role [21,24]. Furthermore, addressing the psychological impact of migraines is critical, as chronic pain conditions like migraines are closely linked with mood disorders such as anxiety and depression. Incorporating mental health support and counseling into treatment plans can significantly improve patient outcomes [1,7].

Patient education is also a key component of individualized care. Studies have shown that understanding the triggers, symptoms, and treatment options for migraines can empower patients to manage their condition more effectively [23]. Moreover, involving patients in decision making helps to tailor treatments according to their preferences, leading to higher adherence and better long-term management [24]. Training programs or workshops designed to improve cross-specialty knowledge—such as recognizing the role of hormonal changes in migraines or understanding the neurological implications of hormonal treatments—are essential for fostering effective collaboration among providers. These initiatives reduce diagnostic delays and mismanagement while empowering patients to access comprehensive care.

As research into the pathophysiology of migraines continues to evolve, there is a need to integrate new therapeutic options, such as CGRP inhibitors, into personalized treatment plans which reflect each patient’s unique needs and circumstances [13,22]. Future efforts should also explore the effectiveness of multidisciplinary care models, identifying the best practices for coordinating treatment across specialties and integrating digital tools to support seamless communication between providers.

## 4. Current and Emerging Treatment Options

### 4.1. Acute Treatments

Acute treatment options for migraines aim to relieve symptoms and restore functionality as quickly as possible. First-line therapies include NSAIDs as well as triptans [5,15]. Triptans are particularly effective for treating moderate-to-severe migraine attacks and work by counteracting the vasodilation and inflammation associated with migraines [22,26]. For menstrual migraines, short-term prophylactic use of triptans starting a few days before menstruation has been shown to be effective in reducing the severity and frequency of attacks [15]. Combination therapies have also shown promise in menstrual migraine management. Studies on a fixed-dose combination of a serotonin receptor agonist and an NSAID demonstrated superior efficacy compared with a placebo for the treatment of menstrual migraine attacks during the mild pain phase. This combination significantly improved the 2 h pain-free response rates, sustained pain relief up to 48 h, and reduced the need for rescue medication while alleviating associated menstrual symptoms. This approach may offer additional benefits by targeting multiple pathways involved in migraine pathophysiology, with a tolerable safety profile [47]. However, it is important to note that while these acute treatments are effective for relieving individual attacks, they do not address the underlying frequency of migraine episodes. Excessive reliance on acute medications, particularly triptans and NSAIDs, can increase the risk of medication-overuse headaches (MOHs), a condition characterized by the worsening of migraine frequency and severity due to frequent medication use. This underscores the need for careful monitoring of acute treatment use and integrating preventive strategies to mitigate the risk of an MOH [48]. Newer acute treatments include CGRP receptor antagonists, which have been found to be effective for patients who do not respond well to triptans or have contraindications to their use [22]. These therapies are promising because they target specific pathways involved in migraine pathophysiology, offering more targeted relief with potentially fewer side effects [23].

### 4.2. Preventive Treatments

Preventive treatments are recommended for patients with frequent, severe, or disabling migraines which significantly impair their quality of life. Traditional preventive therapies, such as beta-blockers, antiepileptics, and tricyclic antidepressants remain effective in reducing migraine frequency and intensity but are often limited by their side effect profiles [5,25]. These limitations have spurred the development of newer options, such as monoclonal antibodies targeting the CGRP pathway—a class of therapies designed to inhibit CGRP activity, a key player in migraine pathophysiology—which offer notable improvements in tolerability and safety compared with traditional agents [13,22]. For patients with chronic migraines, onabotulinumtoxin-A has been proven effective, with benefits in reducing headache days and improving quality of life, particularly in those unresponsive to oral preventives [49].

Recent real-world evidence (RWE) studies have demonstrated the sustained efficacy of monoclonal antibodies targeting the CGRP pathway in reducing migraine frequency, even among patients with complex comorbidities and multiple prior treatment failures. These studies report significant improvements in monthly migraine days, disability scores, and patient-reported outcomes, alongside high responder rates and favorable tolerability profiles. Such findings underscore the integration of these therapies into long-term, personalized migraine management strategies, reflecting their value in both episodic and chronic migraine populations [50]. For women with hormonally triggered migraines, hormone-based preventive strategies, such as continuous or extended-cycle hormonal contraceptives, may stabilize estrogen levels and reduce hormone withdrawal migraines. However, careful assessment is needed, especially for women with MAs, given the increased thrombotic risks associated with these therapies [14,26]. Similarly, HRT may benefit menopausal migraines but must be tailored to individual health risks and migraine characteristics [7,21]. Emerging therapies targeting hormone-sensitive pathways—such as estrogen modulators or melanocortin-4 receptor (MC4R)-targeted treatments—could provide additional options, particularly for postmenopausal women with worsening symptoms due to hormonal changes [44].

For women of childbearing age, preconception counseling is critical to evaluate the teratogenic risks associated with some preventive treatments, such as certain antiepileptic drugs, which are contraindicated during pregnancy. Safer alternatives, including non-pharmacological approaches and emerging therapies, should be discussed as part of a comprehensive care plan. Although CGRP-targeted therapies show promise, further research is needed to confirm their safety in pregnant women [44].

### 4.3. Non-Pharmacological Approaches

Non-pharmacological approaches play a crucial role in the comprehensive management of migraines, especially for individuals who cannot tolerate medications or prefer to minimize drug use. Lifestyle modifications, including regular sleep patterns, balanced diets, hydration, and exercise, are foundational strategies which help manage triggers and improve overall well-being [24]. Behavioral therapies such as cognitive behavioral therapy (CBT) and biofeedback have also shown effectiveness in reducing the frequency and severity of migraine attacks by helping patients manage stress and modify behaviors which may trigger migraines [1,22].

Additionally, complementary and alternative therapies, such as acupuncture, relaxation techniques, mindfulness and yoga, are increasingly being integrated into migraine treatment plans. Studies have demonstrated that mindfulness-based interventions, particularly when combined with standard care, significantly improve headache frequency, quality of life, and medication intake, especially in patients with chronic migraine and medication overuse headache [51]. These approaches provide significant relief, particularly for patients experiencing high levels of stress-related or hormone-triggered migraines [2,15]. Devices that use neuromodulation, such as transcutaneous electrical nerve stimulation (TENS) and non-invasive vagus nerve stimulators, represent another promising area of non-pharmacological treatment. These devices can be used as both acute and preventive therapies, offering patients non-drug options to manage their symptoms [13,26]. Physiotherapy has also emerged as a valuable addition to migraine management, particularly in combination with pharmacological treatments such as onabotulinumtoxin-A. Physiotherapy protocols, including manual therapy and active exercise regimens, have been shown to reduce headache frequency, duration, and intensity by addressing central sensitization and promoting neuroplasticity. These interventions target dysfunctions in both trigeminal and extra-trigeminal areas, enhancing overall migraine-related outcomes [52,53]. As our understanding of migraine pathophysiology deepens, the integration of non-pharmacological and pharmacological approaches continues to be crucial for personalized and effective migraine care.

## 5. Research Gaps

### 5.1. Need for Female-Specific Clinical Trials

While women are well-represented in migraine clinical trials, these studies often fail to stratify participants by reproductive stage (e.g., pregnancy, perimenopause, or menopause) or consider hormonal contraception or other gender-specific factors [22,26,54]. This oversight limits the understanding of how hormonal changes influence migraine patterns and treatment responses, especially given the exclusion of pregnant or lactating women from most pharmacological trials [14]. There is a pressing need for trials which evaluate the safety and efficacy of therapies tailored to women at different life stages, including those addressing teratogenic risks associated with prophylactic medications. Safer strategies, combined with wider adoption of effective contraceptive methods, could reduce risks and improve outcomes globally [13,55]. Recent insights underscore the complexity of hormonal influences on migraines, highlighting a need for future research to delineate the mechanisms underpinning hormone–migraine interactions across different life stages [17].

### 5.2. Technological Advances in Migraine Management

Technological innovation is transforming migraine management through digital health tools, such as wearable devices and smartphone applications, which enable personalized care by monitoring symptoms, identifying triggers, and tracking treatment responses [56]. Neuromodulation devices, including transcranial magnetic stimulation (TMS) and vagus nerve stimulators, provide drug-free alternatives for acute and preventive treatment, showing promising results in reducing migraine severity [23]. Furthermore, artificial intelligence (AI) and machine learning are being developed to analyze patient data and predict migraine triggers, offering potential for highly individualized preventive strategies [15]. Future research should focus on validating these technologies in diverse populations and ensuring their accessibility for all patients.

### 5.3. Education and Awareness

Education is a cornerstone of improving migraine management, addressing stigma and empowering both patients and clinicians. Public awareness campaigns can help shift perceptions of migraines as a serious neurological disorder, while targeted training for healthcare providers can enhance their understanding of gender-specific aspects, including the interplay between hormonal changes and migraines [12,15]. Patient education, particularly regarding triggers, treatment options, and self-management, improves adherence and outcomes. Pain neuroscience education (PNE) has emerged as an innovative approach in managing chronic pain conditions, including migraines. By helping patients understand the neurobiological mechanisms of pain, PNE reduces fear and misconceptions about the condition, which can improve self-management, decrease disability, and enhance function. Additionally, PNE has been shown to alleviate psychosocial factors, encourage movement, and reduce healthcare utilization, making it a valuable tool in migraine education and treatment strategies [57]. Patient support groups and self-help initiatives play a critical role in this context, providing peer support, resources, and guidance to individuals living with migraines. These organizations foster a sense of community and empower patients to take an active role in their care. Additionally, digital platforms and community resources can facilitate the dissemination of comprehensive educational materials, ensuring that both patients and healthcare providers have access to the knowledge needed to effectively address migraines [22,23].

## 6. Future Directions

To advance migraine care, future efforts must integrate the insights from clinical research, technological advancements, and educational initiatives. Validating digital tools and AI-powered interventions in real-world settings will enhance their clinical utility, while public and professional education can combat stigma and improve diagnosis and treatment access. A truly multidisciplinary approach requires the collaboration of neurologists, gynecologists, endocrinologists, primary care providers, pain specialists, mental health professionals, and physiotherapists to address the diverse physiological, hormonal, and psychological dimensions of migraines. Such coordinated care is particularly important for women, whose migraines are often influenced by hormonal fluctuations and life stage transitions. By combining personalized medicine, innovative technologies, and comprehensive education, healthcare systems can deliver holistic and effective management strategies which optimize outcomes for migraine patients worldwide.

## 7. Conclusions

Migraines disproportionately affect women, particularly during their reproductive years, due to the complex interplay of hormonal, neurological, and lifestyle factors. Effective management requires a personalized approach which considers the unique needs of female patients across different life stages. Despite advances in both pharmacological and non-pharmacological treatments, there remain significant gaps in the research, particularly regarding the impact of hormonal changes and the need for tailored therapies during pregnancy and menopause. Future efforts should focus on expanding female-specific clinical trials, leveraging technological innovations, and enhancing education and awareness to improve diagnosis, treatment, and the overall quality of life for women with migraines (Figure 1).

## Figures and Tables

**Figure 1 healthcare-13-00164-f001:**
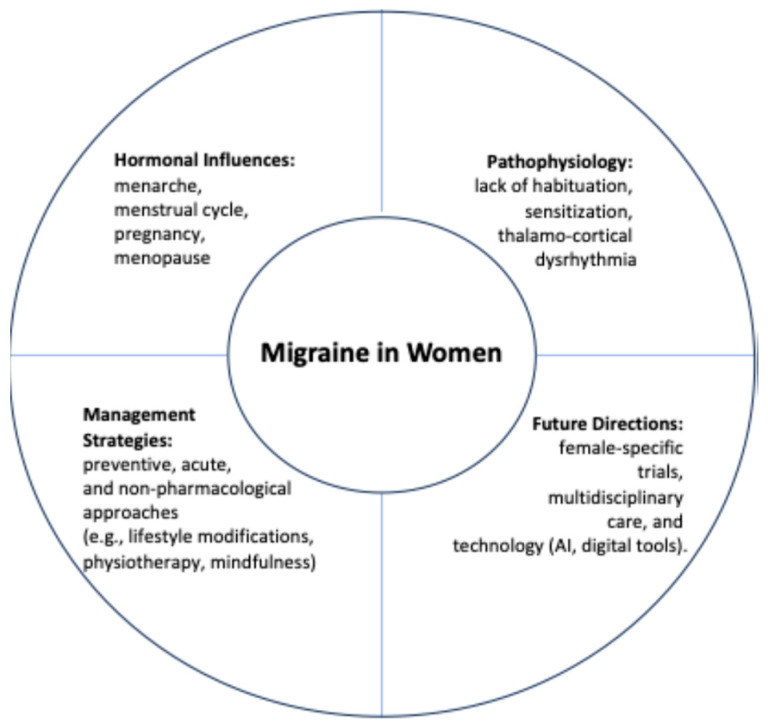
Comprehensive framework for understanding and managing migraines in women. Interplay of hormonal, pathophysiological, and management factors in female migraine care, emphasizing the importance of tailored, multidisciplinary strategies and future research directions.

## Data Availability

No new data were created or analyzed in this study.

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
