# Peer review of "Framing and Management of Migraines in Women: An Expert Opinion on Challenges, Current Approaches, and Future Multidisciplinary Perspectives"

_healthcare, 2025, doi:10.3390/healthcare13020164_

Round 1
Reviewer 1 Report
Comments and Suggestions for Authors
Dear authors,
Thank you for give me the opportunity to revise your interesting manuscript. Migraine represents represents one of the most severe headache in terms of pain intensity and of headache related disability, especially for women. Therefore, a challenge is to find multi-professional new approaches for a more effective management of migraine. In this line your work could be of interest for a broad resercher involved in the study of this complex disease. Please find some specifc suggestions below.
Introduction
- The pathophysiology of migraine should be better introduced in order to highlight the burden that you described related also to their complex physiopathology in which lack of habituation, sesitization, thalamo-cortical dysrethimia determine the cycle of a migraine attack.
4. Current and Emerging Treatment Options
- 4.2. Preventive Treatments. Why did you non mentioned Onoabotulinum toxin A in the preventive treatments?
- 4.3. Non-Pharmacological Approaches. With regard thise kynde of treatments, I think that you should added physiotherapy. Physiotherapy in the last year represents a valid and emerging treatment to support pharmacological treatments, and theri combination with pharmacological treatments could improve the effectiveness in some headache parameters. Please take in to consideration some articles:
Deodato M, Granato A, Ceschin M, Galmonte A, Manganotti P. Algometer Assessment of Pressure Pain Threshold After Onabotulinumtoxin-A and Physical Therapy Treatments in Patients With Chronic Migraine: An Observational Study. Front Pain Res (Lausanne). 2022 Feb 10;3:770397. doi: 10.3389/fpain.2022.770397. PMID: 35295800; PMCID: PMC8915742.
Deodato M, Granato A, Buoite Stella A, Martini M, Marchetti E, Lise I, Galmonte A, Murena L, Manganotti P. Efficacy of a dual task protocol on neurophysiological and clinical outcomes in migraine: a randomized control trial. Neurol Sci. 2024 Aug;45(8):4015-4026. doi: 10.1007/s10072-024-07611-8. Epub 2024 May 29. PMID: 38806882; PMCID: PMC11255006.
an other emerging non-pharmacological treatmens for migraine is mindfulness, please see the suggesting references below
Grazzi L, D'Amico D, Guastafierro E, Demichelis G, Erbetta A, Fedeli D, Nigri A, Ciusani E, Barbara C, Raggi A. Efficacy of mindfulness added to treatment as usual in patients with chronic migraine and medication overuse headache: a phase-III single-blind randomized-controlled trial (the MIND-CM study). J Headache Pain. 2023 Jul 14;24(1):86. doi: 10.1186/s10194-023-01630-0. PMID: 37452281; PMCID: PMC10347788.
- 5.3. Education and Awareness. Pain neuroscince education has emerged as an important way to treat chronic pain, it seems that PNE could reduce pain and improve patient knowledge of pain, improve function and disability, reduce psychosocial factors, enhance movement, and minimize healthcare utilization. I think that a discussion about PNE may make this paragraph more attractive.
6. Future Directions
- "A truly multidisciplinary approach requires the collaboration of neurologists, gynecologists, endocrinologists, primary care providers, pain specialists and mental health professionals to address the diverse physiological, hormonal and psychological dimensions of migraine" in line with the paragraph concerning non-phramacological treatments I suggest you to add also physiotherapists in the professionals involved in migraine multydisciplinary management. In some Europe headache center all this figure work together for a best management and they also have a physiotherapist dedicated and specialized in headache treatment.
Reviewer 2 Report
Comments and Suggestions for Authors
This article is a comprehensive review of migraine in women, from the association with hormonal cycles to non-pharmacological approaches. There are several points that need to be revised for better visibility.
1, Add figures and tables, review the recent review article on similar topic (Kim S, Park JW. Migraines in Women: A Focus on Reproductive Events and Hormonal Milestones. Headache Pain Res. 2024;25(1):3-15.).
2, Define the abbreviation MC4R.
3. Prevention of menstrual migraine with triptans was recommended based on RCT in women. To summarise the main results of trials in menstrual migraine.
Reviewer 3 Report
Comments and Suggestions for Authors
This review provides excelent information value about hormonal aspects of migraine with regards of current treatment options.
I have no remarks to the article
Optional:
In the part 4.1, Acute treatment of migraine, I would prefer to mention the risks of acute treatment only increasing the risk of MOH.
I.e. fixed triptan/NSAID combination is a very powerful tool to alleviate migraine attack, but since these methods dont influence the migraine frequency, they dont alleviate the risk of developing MOH.
